# Development of a Solid-Phase Extraction Method Based on Biocompatible Starch Polyurethane Polymers for GC-MS Analysis of Polybrominated Diphenyl Ethers in Ambient Water Samples

**DOI:** 10.3390/molecules27103253

**Published:** 2022-05-19

**Authors:** Qian Zhang, Chukwunonso P. Okoli

**Affiliations:** 1Institute of Geographic Sciences and Natural Resources Research, Chinese Academy of Sciences, Beijing 100101, China; 2Department of Chemistry, Alex Ekwueme Federal University, Ndufu Alike, Abakaliki 482131, Nigeria

**Keywords:** PBDEs, starch-based polyurethane polymer, SPE, GC-NCI-MS

## Abstract

A new solid-phase extraction (SPE) method for the extraction, enrichment, and analysis of eight polybrominated diphenyl ethers (PBDEs) in water was developed. The current approach involves using a cross-linked starch-based polymer as an extraction adsorbent and determining the PBDE analytes of interest using gas chromatography-mass spectrometry in negative chemical ionization mode (GC-NCI-MS). The starch-based polymer was synthesized by the reaction of soluble starch with 4,4′-methylene-bis-phenyldiisocyanate as a cross-linking agent in dry dimethylformamide. Various parameters impacting extraction efficiencies, such as adsorbent quantity, sample volumes, elution solvents and volumes, and methanol content, were carefully optimized. The 500 mg of starch-based polymer as an adsorbent used to extract 1000 mL of spiked water, presented high extraction recoveries of eight PBDEs. The linearity of the extraction process was investigated in the range of 1–200 ng L^−1^ for BDE-28, 47, 99, 100, and 5–200 ng L^−1^ for BDE-153, 154, 183, and 209, with coefficients of determination (r2) exceeding 0.990 for all PBDEs. The limits of detection (LODs) ranged from 0.06 to 1.42 ng L^−1^ (S/N = 3) and the relative standard deviation values (RSD) were between 3.6 and 9.5 percent (*n* = 5) under optimum conditions. The method was successfully used to analyze river and lake water samples, where it exhibited acceptable recovery values of 71.3 to 104.2%. Considering the excellent analytical performance and comparative cost advantage, we recommend the developed starch-based SPE method for routine extraction and analysis of PBDEs in water media.

## 1. Introduction

Polybrominated diphenyl ethers (PBDEs) are widely utilized in polymers for textiles, electronics, and home furnishings as additive-type flame retardants [1]. Because some brominated flame retardants are not chemically attached to plastic or fabrics, they can be released and accumulated in the environment [2]. Toxicity assessments revealed that PBDE congeners, which have been shown to have endocrine-disrupting properties, may induce liver and thyroid toxicity in wildlife and humans [3]. Though PBDEs have low water solubility, their bioaccumulation and persistence in the environment confers additional toxicological risks. They can enter the human body through the food chain or even by drinking water because they have a strong affinity for particulate materials [4]. PBDEs have received growing attention because of their potential environmental harm, even at trace amounts. As a result, PBDE levels in environmental samples must be monitored effectively and efficiently. In view of this scenario, global and national regulatory agencies have established environmental quality standards (EQS) for PBDEs in the environmental media. The maximum acceptable limit (MAL) for water was 0.14 mg/L for inland water and 0.014 mg/L for surface water. Due to the trace nature of MAL for PBDEs in surface water, an analysis of their levels in water media requires a high level of enrichment as well as sensitive detection methods.

High-performance liquid chromatography (HPLC) [5] and gas chromatography (GC) [6,7,8] were the two most widely used analytical techniques for PBDE in water media. For trace organic contaminants, the extraction, preconcentration, and clean-up of analytes have been the most difficult procedures [9]. One of our earlier investigations [10] used ultrasound-assisted dispersive liquid–liquid microextraction in conjunction with gas chromatography-mass spectrometry (GC-MS) in both negative chemical ionization (NCI-MS) and electron ionization (EI-MS) techniques. Current extraction and concentration techniques for PBDEs from environmental water samples are dependent on the concepts of liquid–liquid extraction (LLE) and solid-phase extraction (SPE) [11]. In LLE, extraction, preconcentration, and clean-up are conducted as separate procedures. Meanwhile, the invention of solid-phase extraction addressed the challenges by combining these three steps into one single procedure in most SPE operations and equally created room for modulating the affinity of the adsorbent phases toward analytes of interest. Aside from the benefits of high recovery, fast extraction, low solvent consumption, and ease of automation/operation, SPE’s ability to tune the analytical selectivity toward analytes of interest has made it the method of choice for pollutants of emerging concern. Based on this, the choice of adsorbent, which controls the affinity toward the analyte of interest, is a critical factor in achieving high enrichment efficiency in SPE.

In view of the foregoing, many classes of materials such as modified silica [12], bamboo charcoal [13], multi-walled carbon nanotubes [14], single-walled carbon nanotubes [10,15], β-cyclodextrin copolymers [5], hydrophilic hyper-cross-linked polymeric materials [16], metal–organic frameworks (MOFs) [7], and various kinds of molecularly-imprinted polymers [17], have been applied as sorbent phases for PBDEs with varying degrees of efficiency. However, the widespread application of most of the reported SPE sorbents suffered from one or more limitations in cost, extraction efficiency, analyte selectivity, narrow extraction spectrum, and non-renewable source [18]. Therefore, there is an obvious need for the development of improved SPE sorbents for PBDEs.

With the obvious advantage of green chemistry strategies, current global attention in materials science is leaning toward environmental friendliness and renewable resources. Aside from its inexpensive and biodegradable nature, starch’s ability to interact physically and chemically with a wide range of molecules made it an ideal precursor for the development of pollutant-targeted adsorbents. Cross-linking starch with appropriate cross-linking agents allows for the introduction of important chemical functionalities, thus promoting the starch polymer’s affinity for certain analytes to be tuned [19]. The tunable affinity of cross-linked starch improves analytical selectivity and makes it an effective solid-phase adsorbent. To remove contaminants from water, a variety of starches and starch derivatives cross-linked with diethylenetriamine [20], acyclic polyamine [21], amino [22], acrylonitrile, ethylenediamine, and other agents [23,24], have been prepared and used as adsorbents. Furthermore, our research group has produced a series of new generations of starch-based adsorbents with remarkable improvements in adsorption properties toward phthalate esters [25], polycyclic aromatic hydrocarbons [26], nitrophenols [27], and tetracycline [28,29], by incorporating desirable functional groups into the starch’s structural backbone. The possession of an acceptable adsorption–desorption profile is a crucial factor in selecting SPE sorbents; the current study seeks to deploy and develop starch-polyurethane polymer as an SPE material for the enrichment of eight PBDEs (BDE-28, 47, 99, 100, 153, 154, 183, and 209) that are commonly found in water media. Various parameters affecting extraction efficiency, such as adsorbent amount, elution solvent, elution volume, sample volume, and methanol content, were optimized.

## 2. Experimental

### 2.1. Reagents and Materials

Mixed standard containing solution of 2.500 mg L^−1^ of all eight PBDE congeners: decabromodiphenyl ether (BDE-209), 2,2′,3,4,4′,5′,6-heptabromodiphenyl ether (BDE-183), 2,2′,4,4′,5,6′-hexbrominated diphenyl ether (BDE-154), 2,2′,4,4′,5,5′-hexbrominated diphenyl ether (BDE-153), 2,2′,4,4′,6-pentabrominated diphenyl ether (BDE-100), 2,2′,4,4′,5-pentabrominated diphenyl ether (BDE-99), 2,2′,4,4′-tetrabrominated diphenyl ether (BDE-47), and 2,4,4′-Tribrominated diphenyl (BDE-28), was purchased from Accustandard (New Haven, CT, USA). Acetone, n-hexane, dichloromethane, and methanol in the HPLC grade were purchased from Aldrich (Milwaukee, WI, USA). Supplied *N*,*N*-dimethylformamide (98%), 4,4′-methylene-bis-phenyldiisocyanate (98%), and analytical reagent-grade soluble starch were sourced from Aladdin Corp (Shanghai, China). All the reagent solutions were prepared with double-distilled water, except where otherwise stated. Six ml capacity polypropylene SPE column (Polypro, 2.CR0006.0001) and polyethylene frits for SPE columns (6 mL PE frits 20 µm, 2.CR06PE.0001) were purchased from CNW Technologies, Shanghai, China.

### 2.2. SPE Procedures

Preparation of the polymer phase was done by a one-step direct reaction (polymerization) of starch with 4,4-methylene bisphenyl diisocyanate (4,4-MDI) as a cross-linking agent as represented in the synthesis scheme (Figure 1). The typical synthesis procedure for the preparation and full characterization results of the starch–polyurethane polymer (SPP) has been described in detail in our previous reports [9,28,30]. The SPE cartridges were assembled by packing aliquots of SPP adsorbent in the 6 mL polypropylene cartridge. Before packing the adsorbent, a polyethylene frit was placed at the base of the cartridge. Thereafter, the aliquot of the adsorbent was added and another polypropylene frit was placed above it to keep the starch polymer adsorbent in place. The adsorbent-packed SPE cartridge was preconditioned with 5 mL methanol to eliminate air and leach impurities before loading water samples, followed by 5 mL ultrapure water to equilibrate the phase. Aliquots of water samples spiked with eight PBDEs were percolated through the cartridge at a rate of 10 mL min^−1^. After that, the cartridge was flushed with 5 mL ultrapure water to eliminate any co-absorbed matrix materials and dried for 10 min under negative pressure. The PBDE analytes adsorbed on the cartridge were then eluted with the suitable solvent. The eluents were percolated in a glass tube and dried on a nitrogen evaporator (Organomation, South Berlin, MA, USA) before being re-dissolved in 1 mL hexane for GC-MS analysis.

All the experiments (SPE extraction and GC analysis) were carried out in triplicate, and the average values were further processed for optimization and validation using OriginPro (OriginLab Corporation, Northampton, MA, USA).

### 2.3. GC-MS Instrumentation

An Agilent 7890 gas chromatograph equipped with a 5975B mass-selective detector operating in negative chemical ionization (NCI) mode, was used to analyze the PBDEs. The GC system was connected to an HP-5ms capillary column (15 m × 0.25 mm I.D., film thickness, 0.25 μm). To increase sensitivity, the mass-selective detector was set to be on the selected ion-monitoring mode (SIM) with NCI, and the two most intense isotope peaks from the mass spectra corresponding to *m*/*z* = 79 and 81 for tri- to hepta-BDEs, and *m*/*z* 79, 81, 486.7, and 488.7 for BDE-209.

All injections were done in spitless mode with volumes of 2 µL in each case, and the injector temperature was kept at 280 °C. The GC oven temperature was set as follows: 100 °C (hold 2 min) to 310 °C at 20 °C min^−1^, final temperature 310 °C (hold 3 min). The MS transfer line was kept at 280 °C, and the quadruple and ion sources were both kept at 150 °C. Methane (99.995%) was utilized as a reagent gas at a pressure of 210^−4^ mbar. Our method utilized methane (99.995%) as reagent gas at a pressure of 2 × 10^−4^ mbar and helium (99.999%) as carrier gas at a constant flow rate of 1.0 mL min^−1^. 

### 2.4. Optimization of Relevant SPE Parameters

The amount of SPE adsorbent, nature of elution solvent, eluent volume, water sample volume, nature and amount of sample modifier, as well as nature of the adsorbent phase, are the major determinants of analyte recovery in the SPE process. In view of this, the current study deployed the following standard procedures for optimization of the influential parameters. Optimization experiments were conducted in triplicate. 

### 2.5. Quality Assurance/Quality Control Measures and Water Samples Analysis

Several quality assurance/quality control tests were conducted to compare the developed method with EPA method 1614. Blanks were included in each batch of extractions and instrumental analysis throughout the experiment, to check the methodic error and cross-contamination. Linearity and linear range of the calibration were evaluated. Spiked sample analyses were conducted to evaluate the precision and recovery.

PBDEs spiked double-distilled water (0.05 μg L^−1^) samples were utilized for optimization of influential parameters, whereas real water samples (collected from lake and river in Beijing, China) were used for method performance assessment and validation. All environmental water samples were passed through a 0.45 μL PTFE syringe filter and kept in brown glass containers in the refrigerator at 4 °C to prevent possible microbial degradation of the PBDE analyte of interest prior to instrumental analysis.

## 3. Results and Discussion

### 3.1. Adsorbent Design and Adsorbent-Analyte Interaction Mechanism

The design of SPE adsorbents is done in such a way as to enhance effectiveness, efficiency, selectivity, and in some cases, target specific. The effectiveness and efficiency are achieved by creating a strong affinity between the adsorbent and analyte of interest, whereas selectivity is achieved by making the created affinity to be selective amid other coexisting analytes/pollutants [31]. In creating adsorbent–analyte interaction affinity, the chemical functionalities of the analyte of interest are considered since it influences the potential adsorption/retention interaction that controls analyte separation. For PBDEs, the bromo- and phenyl-functional groups are the most prominent. As stated earlier, the choice of starch as the substrate for the adsorbent design is based on its ability to easily react with a wide range of molecules. The adsorbent design for this work, therefore, utilized 4,4-MDI as a cross-linking agent to introduce some key desirable functional groups such as aromatic and amide groups in addition to the hydroxyl functional groups of starch as shown in the synthesis scheme (Figure 1). In view of these prominent functional groups, the starch polymer adsorbent–PBDE interaction is likely dominated by lone pair—lone pair, π-π, and hydrogen bond interactions as shown in Figure 2. These interactions created suitable affinity and selectivity for an effective extraction process. Further improvements in selectivity are achieved with the choice of elution solvent and optimization of other SPE extraction parameters.

### 3.2. Optimization of the SPE Procedure

Before optimizing the SPE parameters, the suitability of the instrumental method (in this case, GC-NCI-MS) was tested by running a PBDE-spiked sample. The obtained GC-NCI-MS chromatogram (Figure 1) of eight PBDE analytes of interest, indicated that with the chosen column condition (pressure and temperature program) and detection mode, the MS detector achieved a very good peak separation, even at the spiked level of 0.05 µg L^−1^.

The optimization experiments showed that the analytical recoveries of PBDEs in the SPE process were influenced by the investigated factors; the amount of SPE adsorbent, the types of eluents (elution solvents), eluent volumes, sample volumes, sample modifiers (methanol content), and the nature of adsorbents. This observation is in accordance with the basic principles of the SPE process as optimization is crucial to understanding the simultaneous interaction of the influential parameters to enable the operator to arrive at the optimum parameter values. Considering the fact that the values of many SPE optimum parameters are influenced by the level (concentration range) of the analyte of interest in the water samples, it has become obvious that the optimization process is a fundamental stage in SPE method development.

#### 3.2.1. Effect of the Amount of Adsorbent

Varying amounts of starch-based polymer adsorbents in SPE devices in the range of 250, 500, 650, and 750 mg were evaluated by extracting 1000 mL water samples. The results in Figure 2 show that as the amount of adsorbent increased from 250 to 500 mg, the extraction recoveries increased. Beyond 500 mg, it was observed that further increments in adsorbent mass to 650 and 750 mg did not result in significant improvements in analytical recoveries. This trend could be interpreted on the basis that 500 mg of the adsorbent was enough to adsorb all the PBDEs available in the selected sample volume. Thus, further increments in adsorbent mass would not result in further adsorption of PBDEs. Therefore, a weight of 500 mg of cross-linked starch-based polymer adsorbent seemed to be sufficient to extract the analytes effectively.

#### 3.2.2. Effect of the Elution Solvent and Elution Volume

The nature and chemistry of the organic solvent used as an eluent have a significant impact on elution efficiency, which is a measure of analyte desorption from the solid phase. Hence, different organic solvents would show different elution power. In view of this, our experiment evaluated five solvents: dichloromethane, acetone, methanol, a mixture of acetone/n-hexane (50:50, *v*/*v*), and dichloromethane/n-hexane (50:50, *v*/*v*). It can be observed from the results shown in Figure 3, that the dichloromethane/n-hexane mixture had the highest extraction recovery (70.8–105.3%) compared with dichloromethane (60.2–101.2%), the mixture of acetone/n-hexane (58.7–90.2%), and acetone (58.2–86.2%). The methanol solvent had the lowest extraction recovery (29.4–64.5%). It was also observed that the dichloromethane/n-hexane eluent produced a cleaner extract than the other solvents/solvent mixtures. Oftentimes, the elution efficiency is governed by the polarity index of the solvent. Hence, the observed elution efficiency is in the ascending order of polarity viz: 50:50 dichloromethane/n-hexane (1.6), acetone/n-hexane (2.6), dichloromethane (3.1), whereas acetone and methanol with the same polarity index of 5.1 are expected to exhibit similar elution behaviors. In addition, the extent of solvent interaction with the SPE phase may likely alter the trend due to the polarity index. As a result, the solvent mixture of dichloromethane/n-hexane, which exhibited the greatest performance was chosen as the ideal extraction solvent for subsequent trials.

There is no doubt that the volume of elution solvent affects the elution efficiency [9] Thus, several aliquots of eluent volumes (5–15 mL) were tested to study the influence of this parameter. From the results shown in Figure 4, the extraction recoveries of the eight PBDE congeners improved with the increase in the eluent volume from 5 to 10 mL. However, there was a minimal increase in the recovery when the volume of eluent was increased from 10 to 15 mL. Throughout the experimental procedure, 10 mL dichloromethane/n-hexane (50:50, *v/v*) was employed as the eluent volume. There was no substantial increase in the analytical recoveries beyond the 10 mL volume of dichloromethane/n-hexane (50:50, *v*/*v*). Therefore, 10 mL dichloromethane/n-hexane (50:50, *v*/*v*) was selected as the eluent for further experiments.

#### 3.2.3. Effect of Sample Volume

In order to obtain acceptable analytical results and high concentration factors, five volume levels 250, 500, 750, 1000, and 1500 mL were employed, and the concentration of each spiked analyte was fixed at 0.05 μg L^−1^. The results (Figure 5) showed that the recoveries of the eight target compounds remained consistent in the 250–1000 mL range. However, when the sample volume was increased to 1500 mL, the recoveries of a few congeners, such as BDE-183 and BDE-209, decreased slightly. The enrichment factors for the eight PBDEs were the highest by extracting analytes from the 1000 mL aqueous solution, and the enrichment factors for BDE-28, 47, 99, 100, 153, 154, 183, and 209 were 1017, 906, 916, 941, 852, 792, 786, and 707 folds, respectively. A sample volume of 1000 mL was chosen as the best sample volume based on the enrichment factors.

#### 3.2.4. Effect of Methanol Content and Salt Concentration

To assess the influence of the sample modifier, varying percentages of methanol ranging from 0 to 20% were introduced to a spiked water sample and the experimental results are shown in Figure 6. The observed trend showed that the extraction recoveries have a positive correlation with the methanol content, which was especially noticeable for compounds with a higher number of bromine atoms, such as hexa-, hepta-, and deca-BDEs. The observation was probably due to the increment in the PBDEs’ solubility in the solvent, which equally reduces the number of PBDEs adsorbing on the walls of the sample containers as the methanol content of the samples increased. The highest recoveries were achieved at a methanol content of 15% and no additional variability above the content of 15% was shown. Therefore, this content was used in subsequent experiments. 

The effect of increasing the ionic strength of the water sample was also investigated by adding sodium chloride to water samples spiked with PBDEs at a concentration of 0.05 μg L^−1^. The results showed that the effect of salinity on the recoveries of target compounds was negligible with the concentration of sodium chloride ranging from 0–10% (*w*/*v*). As the ionic strength of real environmental water samples varies from one location to another, it is desirable for salinity to have a negligible influence on analytical recovery. This will also save operational costs (of sodium chloride) and energy (required in sample modulation). Thus, the negligible influence of salinity on the developed SPP material is adjudged as a strong comparative advantage over other materials. So, sodium chloride was not applied in subsequent experiments. 

### 3.3. Comparison of the Efficiency of the Developed Starch-Based Polymer with Commercially Available SPE Adsorbents

To compare the extraction efficiency of the starch-based polymer, Oasis^®^ HLB cartridges (Waters, Milford, MA, USA) were selected for the PBDEs’ extraction. The Oasis^®^ HLB cartridge was made with 200 mg of hydrophilic-lipophilic balanced copolymer sorbent in a 6 mL evacuate polypropylene cartridge, and the SPP cartridge was prepared with 500 mg of SPP adsorbent, which was used to extract 1000 mL water samples with spiked standard solutions at concentrations of 5.0 ng L^−1^. The SPE procedures and GC-MS detection method were the same for the two SPE cartridges. Table 1 shows the recovery and RSD of PBDEs for different SPE cartridges, and we also compared these results with previous reports such as the LC-18 cartridge (Supelco, Bellefonte, PA, USA). The starch-based polymer-packed SPE cartridge displays slightly higher recoveries and better precision than other commercial cartridges. The observation is obviously related to the high enrichment of the starch-based polymer, which is a consequence of the successful incorporation of desirable functional groups into the structural backbone of starch. Though the reviewed LLE methods [32,33] showed comparable performance with the starch polymer SPE, the obvious advantages of SPE over LLE would make the developed SPE a method of choice.

### 3.4. Validation of the Starch-Based Polymer SPE Method

To estimate the accuracy and precision of the starch-based polymer SPE technique, the linearity, coefficients of determination, detection limits, and repeatability were evaluated using spiked samples under optimum conditions. The results for these parameters are shown in Table 2. Linearity was evaluated in the range of 1–200 ng L^−1^ for BDE-28, 47, 99, 100, and 5–200 ng L^−1^ for BDE-153,154,183, and 209. The coefficients of determination (r^2^) ranged from 0.9903 to 0.9986. The analytical precision, which is expressed as relative standard deviation (RSD), was generally less than 10% (*n* = 5). The limits of detection (LODs) were assessed based on a signal-to-noise ratio (S/N) of 3, and the values were from 0.06 to 1.42 ng L^−1^. These results demonstrated that this approach had good extraction recovery for PBDEs in water samples.

### 3.5. Application to Environmental Water

The recoveries of the eight PBDEs from spiked environmental water samples were determined to illustrate the applicability and reliability of the developed extraction method. Before being spiked, the lake and river environmental water samples were tested to confirm that they were free of PBDE contamination. The recovery study was conducted by determining the level of PBDEs in un-spiked and spiked water samples, and the percentage recovery was estimated via the following equation:(1)$$\%\;Recovery=\frac{C_{SP}-C_{UP}}{SL}\times \frac{100\%}{1}$$
where $C_{SP}$ is the PBDEs level of the spiked sample, $C_{UP}$ is the PBDE level of the unspiked sample, and *SL* is the spike level.

Under the optimal conditions as determined in the optimization experiment, the analytical performance of the developed method was evaluated in terms of precision (RSD) obtained by spiked standard solutions at concentrations of 5.0 and 50.0 ng L^−1^. The RSD values were computed using five replicate runs and the recoveries of river water and lake water samples were averaged from three replicate runs, as shown in Table 3. The mean recoveries for all the PBDEs were 67.5–102.4% and 71.2–104.1% at the different concentrations in river water and lake water, respectively. The obtained values showed that the developed polymer phase is an excellent SPE adsorbent for the extraction of trace levels of PBDEs in environmental water samples.

## 4. Conclusions

An efficient and reliable method was developed for the extraction, enrichment, and analysis of trace levels of PBDEs in water samples using a starch-based polymer as an SPE adsorbent coupled with GC-NCI-MS. This current development promotes the concept of a green chemistry approach as the adsorbent was synthesized from a renewable resource. The method was successfully validated with acceptable precision, limit of detection, linearity, and linear range. The results of the comparative performance tests showed that the developed SPE phase is slightly better than the commercially available SPE cartridges. The environmental applicability tests showed that the recoveries of the eight PBDEs in the real environmental water samples (river and lake water samples) were very good even at two different spiked levels. Overall, the application of a developed starch-polyurethane polymer as an SPE adsorbent phase, possesses remarkable merits for the extraction of PBDEs and offers an alternative for routine SPE materials. Considering the comparative cost advantages, we recommend the developed SPE method for the routine extraction and analysis of PBDEs in aqueous media.

## Data Availability

Not applicable.

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
