# Peer review of "Development of a Solid-Phase Extraction Method Based on Biocompatible Starch Polyurethane Polymers for GC-MS Analysis of Polybrominated Diphenyl Ethers in Ambient Water Samples"

_molecules, 2022, doi:10.3390/molecules27103253_

Round 1

Reviewer 1 Report

A very well written introduction to the subject. A smooth transition from the importance of the research goal to human health, the source PBDE of pollution and the effectiveness of the methods currently used for treatment.

line 77 "ideal precursor for .." - I would suggest replacing the word precursor with something else in reference to starch as a potential absorbent

entire chapter 2.1. Reagents and materials - should be removed, and the content of materials or reagents used for the purpose of the research - included in the subsections describing the research methodology in which they were necessary to conduct

line 108-112- the assumption is that the materials used or the methods used should be described accurately enough for the potential reader to recreate the experience. Not all platforms / publications for international readers are available or free of charge, making them unreachable and therefore not repeatable or discussed and referenced.

In the materials and methods (2) chapter, there is no information on the statistical processing of the results.

The work, apart from the small remarks (about which above), is very well written, the selected methods necessary to achieve the goal are correctly selected. The conclusions are preceded by the obtained test results.

Author Response

The authors wish to thank the editor and this reviewer for giving us the opportunity to revise and resubmit our manuscript in this revered journal. We equally thank this reviewers for the painstaking review process, useful comments and constructive recommendations, on the content, structure and language of our paper, which helped us to improve our manuscript.

Based on these comments and suggestions, we have significantly modified the original version of the manuscript. All the reviewers’ concerns have been specially noted and addressed, and major revisions have been carried out according to the reviewers’ suggestions. The authors equally addressed other ancillary reviewers’ questions and provided explanations/rebuttal in very few occasions where the reviewer's recommendations were not fully accepted/complied with.

The detailed revisions are listed below point by point:

Reviewer 1

Response to reviewers’ comment

  1. line 77 "ideal precursor for .." - I would suggest replacing the word precursor with something else in reference to starch as a potential absorbent

Response

The authors thank this reviewer for this comment. Starch was actually deployed as the major starting material for the synthesis of the polymer adsorbent, and as such, cannot be considered as a potential adsorbent. Rather, the word “precursor” or its contextual synonym “substrate” fit into the context being described in the passage. However, the authors believe that “precursor” conveys the meaning better than “substrate”, and therefore appeal to this reviewer to positively consider our explanation.

  1. entire chapter 2.1. Reagents and materials - should be removed, and the content of materials or reagents used for the purpose of the research - included in the subsections describing the research methodology in which they were necessary to conduct

Response

The authors appreciate this reviewer for the observation, which the authors are willing to adopt. However, the journal template for this kind of work requires that the “Material” subsection be detached from the “Experimental procedure” subsection. This style is consisted with most SPE work published with this journal (Molecules). This reviewer can confirm this assertion from the from https://www.mdpi.com/1420-3049/26/20/6163 and other articles in the journal.

  1. line 108-112- the assumption is that the materials used or the methods used should be described accurately enough for the potential reader to recreate the experience. Not all platforms / publications for international readers are available or free of charge, making them unreachable and therefore not repeatable or discussed and referenced.

Response

The authors agree with the reviewer on the need to give procedural details enough to enable the repetition of the work elsewhere. This is the reason the authors went ahead to describe the SPE packing and experimental procedure despite the references already cited. In view of this comment, the authors have added additional details on the synthesis of the polymer phase (see page 3, line 113 to 116 of the revised manuscript).

  1. In the materials and methods (2) chapter, there is no information on the statistical processing of the results.

Response

The authors appreciate this reviewer for this observation. The statistical processing of the results of the work being reported in this manuscript has now been included in this revised version of the manuscript. Please see page 3, line 131 to 133 of the revised manuscript.

Reviewer 2 Report

Qian Zhang and Chukwunonso P Okoli reported on the use of starch-based polymers as SPE for the preconcentration of a series of PBDEs.

Overall, the present manuscript did not present in a way which could attract interests of the readers, as the work was rather based on routine and is produced with the similar concept of a huge number of well-established works previously published in the literature on SPE for the determination of traced toxic chemicals in water-based systems. Although the written English is fine, and the data sets and proper analysis are well organized. I could not accept it in the present form and an immense revision is highly desirable. If the authors could not accord soundly to the following comments, the rejection is possible.

My comments are as follows:

  • The reasonable concept of using such starch-based for polymers as SPE for the extraction of PBDEs is not clear. Why is this type of SPE (starch-based polymers) suitable for the extraction of these compounds in terms of chemistry, regardless of only the green concept of the material synthesis? This is a must (please see this paper which reported the use of starch-based SPE as an example and it should be cited accordingly similar work) : https://www.mdpi.com/1420-3049/26/20/6163

For example, chemical functionalities of the SPE interact well with the PBDEs? Or with what other reasonable reason for the materials of choice? The characterization of the SPEs in this work must be performed with suitable techniques (SEM, FTIR, etc)

  • The selectivity is considered novel and could make the manuscript sound interesting, but based on over all studies of the work, the selectivity is not mentioned at all, and even a single try of using other molecules or chemicals interferences with similar structures to PBDEs, or with the same functionalities should be done.
  • Schematic representation of the starch based SPE should be presented as there is no even single picture illustrating how the SPE was produced and how their chemical structures of functionalities look like. The scheme might make the readers get to the point and find the work earlier to understand.
  • Similarly, the schematic representation of how the SPE interacted with the analytes must be shown.
  • The comparison of the performance was done for both the validation of the starch-based SPE and the application to the real water systems. However, only the commercial SPE were used to for this comparison purpose. More references from the literature for the comparison of the analysis performance is highly required. This is not limited to only SPEs for the PBDEs analysis or preconcentration, but other techniques like liquid-liquid phase extraction should also be included (the Table with up to 7-10 references might help to make this work more interesting).

Author Response

The authors wish to thank this reviewers for giving us the opportunity to revise and resubmit our manuscript in this revered journal. We equally thank the reviewer for the thorough review process, useful comments and constructive recommendations, on the content, structure and language of our paper, which helped us to improve our manuscript.

Based on these comments and suggestions, we have significantly modified the original version of the manuscript. All the reviewer’s concerns have been specially noted and addressed, and major revisions have been carried out in line with the reviewer’s suggestions. The authors equally addressed other ancillary reviewers’ questions and provided explanations/rebuttal in very few occasions where the reviewers recommendations were not fully accepted/complied with.

The detailed revisions are listed below point by point:

  1. The reasonable concept of using such starch-based for polymers as SPE for the extraction of PBDEs is not clear. Why is this type of SPE (starch-based polymers) suitable for the extraction of these compounds in terms of chemistry, regardless of only the green concept of the material synthesis? This is a must (please see this paper which reported the use of starch-based SPE as an example and it should be cited accordingly similar work) : https://www.mdpi.com/1420-3049/26/20/6163

For example, chemical functionalities of the SPE interact well with the PBDEs? Or with what other reasonable reason for the materials of choice? The characterization of the SPEs in this work must be performed with suitable techniques (SEM, FTIR, etc)

Response

The authors do completely agree with this reviewer on the above suggestion. Besides the green concept of material synthesis and biocompatibility of the starch-based polymer, the tunable affinity of cross-linked starch polymer adsorbents makes them excellent for SPE adsorbents for extraction of trace pollutants. The referred paper was useful in revising the manuscript in accordance with the reviewer’s suggestion, and has been cited in the revised version of the manuscript (see line 78 to 82).

  1. The selectivity is considered novel and could make the manuscript sound interesting, but based on over all studies of the work, the selectivity is not mentioned at all, and even a single try of using other molecules or chemicals interferences with similar structures to PBDEs, or with the same functionalities should be done.

Response

The authors thank this reviewer for this important observation and suggestion. The selectivity of the crosslinked starch adsorbents towards analytes of interest is one of the key selling points of this work. In view of this observation, the tunable selectivity concept of crosslinked starch polymer has now been included in the revised manuscript (see line 80 to 82)

  1. Schematic representation of the starch based SPE should be presented as there is no even single picture illustrating how the SPE was produced and how their chemical structures of functionalities look like. The scheme might make the readers get to the point and find the work earlier to understand.

Response

The authors appreciate this reviewer for this comment. Schematic representation of the starch polymer synthesis, which showed the prominent functional groups of the starch polyurethane polymer, has now been included in page 4, line 150 (Scheme 1) of the revised manuscript. Though the graphical abstract was not included as the integral part of the reviewed manuscript, the reviewer’s suggestion has also been captured in the graphical abstract of the manuscript. In view of this suggestion, the authors have now included the graphical abstract as an integral part of the manuscript.

  1. Similarly, the schematic representation of how the SPE interacted with the analytes must be shown.

Response

The authors really appreciate this reviewer for this suggestion, as the importance of the interaction schematics cannot be over emphasized. The schematic representation of how the starch polyurethane polymer SPE adsorbent interacted with the polybrominated diphenyl esters has been included in page 6, line 186 (Scheme 2) of the revised version of the manuscript.

  1. The comparison of the performance was done for both the validation of the starch-based SPE and the application to the real water systems. However, only the commercial SPE were used to for this comparison purpose. More references from the literature for the comparison of the analysis performance is highly required. This is not limited to only SPEs for the PBDEs analysis or preconcentration, but other techniques like liquid-liquid phase extraction should also be included (the Table with up to 7-10 references might help to make this work more interesting).

Response

The authors of the manuscript thank this reviewer for the thorough review of our manuscript. The authors have reviewed more published work and added more comparative data to Table 1 of the revised manuscript as requested by the author.

Reviewer 3 Report

The paper talks about the SPE method for PBDEs, although the data is interesting, the authors have not discussed the QA/QC aspects of the entire process which is very important. Please refer to EPA method 1614 in regards to QA/QC aspects required.

Author Response

The authors wish to thank the editor and reviewers for giving us the opportunity to revise and resubmit our manuscript in your revered journal. We equally thank the reviewers for their painstaking review process, useful comments and constructive recommendations, on the content, structure and language of our paper, which helped us to improve our manuscript.

Based on these comments and suggestions, we have significantly modified the original version of the manuscript. All the reviewers’ concerns have been specially noted and addressed, and major revisions have been carried out according to the reviewers’ suggestions. The authors equally addressed other ancillary reviewers’ questions and provided explanations/rebuttal in very few occasions where the reviewers recommendations were not fully accepted/complied with.

The detailed revisions are listed below point by point:

The paper talks about the SPE method for PBDEs, although the data is interesting, the authors have not discussed the QA/QC aspects of the entire process which is very important. Please refer to EPA method 1614 in regards to QA/QC aspects required.

Response

Though the results of the quality control tests were implicitly discussed in the manuscript, the authors have now added a full paragraph that described the quality assurance/quality control aspect of the study in page 5, line 157 to 162 of the revised version of the manuscript.

Round 2

Reviewer 2 Report

The authors have considerable revised the manuscript. The huge improvement has been done. All the comments have soundly been reponded. The manuscript can be accepted for publication in Molecules.

Reviewer 3 Report

Can be accepted.